# Physics Informed Machine Learning with Misspecified Priors:
## An analysis of Turning Operation in Lathe Machines

**Zhiyuan Zhao, Xueying Ding, Gopaljee Atulya, Alex Davis, Aarti Singh**

Carnegie Mellon University, 5000 Forbes Ave, Pittsburgh, PA 15213

### Abstract

The recent development of physics informed neural networks (PINNs) has explored the inclusion of prior physics knowledge into the objective function of deep learning models as differential equation loss component to supervise learning of complex systems under data-constrained settings. However, PINN framework requires that expert-provided knowledge about the physical system is perfectly accurate, neglecting cases where there is potential for fallibility in expert judgment. We extend this research to consider the effect of explicit fallible expert judgment in the learning process. First, we theoretically upper bound the effect of fallible expert-provided information on the convergence of PINNs to the true solution. We show how to opportunistically leverage fallible expert knowledge when data are scarce, and gracefully diminish reliance on inaccurate expert judgment as more data are acquired. Second, we examine the limitations of the PINN in learning noisy real-world physical systems, and apply a modified Seq2seq learning with applications in turning operation in lathe machines. We also propose a combination of PINN framework with recurrent neural networks for predicting system behavior outside the training domain.

## Introduction

In recent years, experts, scientists, engineers and technicians have provided valuable prior knowledge to accelerate learning and optimization for AI systems. For example, research in rejection learning [4, 3] and learning-to-defer [14], train a model to learn when to query an expert to make a classification [11, 20]. In predictive maintenance and manufacturing processes, combination of structured knowledge from experts with machine learning (ML) has proven to reduce costs and optimize machine operation [9, 1, 12]. Traditionally, the technicians or engineers can read off patterns in measurable quantities (e.g., sound, vibration, force) during machine operation, that can capture the error modes due to degradation, corrosion, overheating, etc. Physics knowledge in the form of ordinary differential equations (ODEs) or partial differential equations (PDEs) can also be regarded as structural knowledge. With the introduction of physics-informed machine learning [10, 21, 23, 19, 5, 13, 15, 18], the expert's knowledge of underlying physics can be incorporated into deep learning for predictive maintenance.

The current physics-informed approaches refer to simulated data-sets with well-known physical functions and boundary conditions that accurately describe the phenomenon being studied. In real-world noisy systems an expert's assessment of the underlying physics may never be perfectly accurate. Thus, a method that can flexibly combine expert physical knowledge with data-driven models makes it possible to obtain the best of both worlds: small-data accuracy by leveraging expert judgement and large-data generalizability despite potentially misspecified expert knowledge.

In this paper we take physics-informed neural network (PINN) under the scenario where experts give misspecified judgement of machine behavior. Our main contributions are:

- A theoretical proof of convergence of PINN under misspecification, and show how to diminish the reliance on inaccurate physics knowledge for PINN with empirical evaluation on chattering data.

- We introduce a combination of PINN with recurrent neural network to tackle the problem of imbalanced data distribution, which is a more common scenario in the time series prediction for predictive maintenance.

- We examine current limitations of PINN, and apply modified seq2seq training resulting in faster convergence.

## Background: Introduction to PINN

Physics-informed neural networks (PINNs), introduced by Raissi *et al*. [15], are effective tools for solving differential equations. This approach leverages the fact that partial derivatives are easily calculable from neural networks using auto-differentiation. As a result, the neural network tries to minimize the linear combination of structural loss and empirical data loss. Here we refer to the simplest form of PINN, which solves the following differential equation problem:

Let $\Omega$ be an open bounded set in $\mathbb{R}^d$, with a boundary $\partial\Omega$. A partial differential equation takes the form: $\mathcal{L}[u](x) = f(x), \forall x \in \Omega, \mathcal{B}[u] = g(x), \forall x \in \partial\Omega$, where $\mathcal{L}$ is a differential operator and $\mathcal{B}$ is a boundary condition (such as Dirichlet or Neumann). For simplicity, consider the linear elliptic PDE with Dirichlet boundary condition and thus, the boundary condition becomes: $u(x) = g(x), \forall x \in \partial\Omega$.

Given a class of neural networks $\mathcal{H}$, the goal is to find a neural network $h$ that best approximate the solution $u$ of the above equation, which is equivalent to minimize

PINN loss [15] defined as: $Loss^{PINN}(x_r, x_b, h, \lambda_r, \lambda_b) = \lambda_r \|\mathcal{L}[h](x_r) - f(x_r)\|^2 + \lambda_b \|h(x_b) - g(x_b)\|^2$, where $x_r, x_b$ are points from $\Omega$ and $\partial\Omega$ respectively. We consider the setting where there are also a set of collected measurement points (e.g. sensor measurements on a lathe machine) $(x, u(x)), x \in \Omega$. Then, treating the physics prior as regularization, the goal is to minimize a Data-Regularized PINN (DRP) loss:

$$Loss^{DRP}(x_r, x_b, x_t, h, \lambda_r, \lambda_b) = \lambda_r \|\mathcal{L}[h](x_r) - f(x_r)\|^2$$
$$+ \lambda_b \|h(x_b) - g(x_b)\|^2 + \|h(x_t) - u(x_t)\|^2 \tag{1}$$

Minimizing Data-Regularized PINN loss will also simultaneously lead to minimize the PINN Loss, because $Loss^{PINN} \leq Loss^{DRP}$, and $\|h(x_t) - u(x_t)\|^2$ will be minimized so that $h$ approximates $u$, while the empirical data can be noisy with some i.i.d noise; i.e. $u(x_t) + \epsilon$.

## Convergence Under Misspecification

Despite prior works indicating that optimizing the empirical PINN loss will result in an optimal solution of the ODE/PDE [17], a more common scenario is that experts can only provide limited physical information. The desired solution might be described by a PDE that is similar in behavior, but different in coefficients, boundary conditions, etc. from the expert's guidelines. Thus, the physics priors are actually misspecified, so the resulting PDE is:

$$(\mathcal{L} + \epsilon_{\mathcal{L}})u(x) = f(x) + \epsilon_f(x), \; \forall x \in \Omega$$
$$u(x) = g(x) + \epsilon_g(x), \; \forall x \in \partial\Omega \tag{2}$$

Consider misspecified data-regularized PINN (MDRP) loss:

$$Loss^{MDRP}(x_r, x_b, x_t, h, \gamma) = \|h(x_t) - u(x_t)\|^2$$
$$+ \lambda_r \|(\mathcal{L} + \epsilon_{\mathcal{L}})[h](x_r) - f(x_r) - \epsilon_f(x_r)\|^2 \tag{3}$$
$$+ \lambda_b \|h(x_b) - g(x_b) - \epsilon_g(x_b)\|^2$$

To analyze how minimizing the empirical misspecified data-regularized PINN loss can help to estimate a solution of the Equation (2) with misspecified experts' guidance, we first formulate a simpler scenario where the both the PDE and neural networks satisfy the following:

- **Smoothness.** The function we are estimating, $u \in L^2(\Omega)$, should satisfy the Hölder continuity conditions. In addition, operator $\mathcal{L}, \mathcal{B}$, and functions $f, g, h, u$ are in $\alpha$ Hölder space for some $0 < \alpha < 1$.
- **Neural Network approximation.** There exist sufficiently large neural networks that are able to approximate the solutions to the PDE to arbitrary accuracy.
- **Ellpticity.** The operator $\mathcal{L}$ is linear and elliptic.

The following Lemma bounds the data-regularized PINN loss using the empirical misspecified data-regularized PINN loss on which the neural network is trained.

**Lemma 1.1** Suppose $m_r$ and $m_b$ denotes the number of collocation points where the differential equation is enforced $\tau_r = \{x_r^i\}_{i=1}^{m_r}$ and $\tau_b = \{x_b^i\}_{i=1}^{m_b}$ are the boundary points.

Further, assume that for any $x_r \in \Omega$ and $x_b \in \partial\Omega$, there exists $x_r' \in \tau_r$ and $x_b' \in \tau_b$ such that $\|x_r - x_r'\| \leq \epsilon_r$ and $\|x_b - x_b'\| \leq \epsilon_b$. In addition, let $m_t$ be the number of measurement data and $\tau_t = \{x_t^i\}_{i=1}^{m_t}$ be the set of samples from measurement. Also assume that for any $x \in \Omega$, there exists $x_t' \in \tau_t$ such that $\|x_t - x_t'\| \leq \epsilon_t$. Then we can bound $Loss^{PINN}$ or $Loss^{DRP}$ above with empirical average $Loss_m^{MDRP}$:

$$Loss^{DRP} \leq C_m Loss_m^{MDRP}(h, \lambda') + \epsilon_h + \epsilon_m \tag{4}$$

where $C_m = 3\max\{C_r m_r \epsilon_r^d, C_b m_b \epsilon_b^{d-1}, C_t m_t \epsilon_t^d\}$, $\lambda' = 3\lambda$, and $\epsilon_h, \epsilon_m$ are error terms with Hölder constants and misspecifications, respectively.

**Theorem 1** Suppose for each $m_t$, $\mathcal{H}_m$ contains a network $h_m$ that perfectly optimizes the empirical misspecified data-regularized PINN loss, that is, $Loss_m^{MDRP}(h_m) = 0$. Also, let $m_r = O(m_t)$ and $m_b = O(m_t^{\frac{d-1}{d}})$. Then with probability at least $1 - \delta$ for any $\delta \in (0,1)$, over iid samples:

$$Loss^{DRP}(x_r, x_b, x_t, h_m, \lambda) \leq \mathcal{O}(m_t^{-(\frac{2\alpha}{d}+1)} \log m_t)$$
$$+ \mathcal{O}(m_t^{-\frac{2\alpha}{d}} \log m_t) + \mathcal{O}(m_t^{-1}) \tag{5}$$

where we pick specific regularization parameter $\lambda = \mathcal{O}(m_t^{-1})$ so that the misspecification error will be $\mathcal{O}(m_t^{-1})$ in Eq. (5), i.e. the expert-provided misspecified physics knowledge is washed away as more measurement data is collected. Full proof of *Lemma 1.1* and *Theorem 1* is detailed in Appendix A.

Noticing that $Loss^{PINN}$ is upper bounded by $Loss^{DRP}$, then with probability 1 over iid samples:

$$\lim_{m_t \to \infty} Loss^{PINN}(x_r, x_b, x_t, h_m, \lambda') =$$
$$\lim_{m_t \to \infty} Loss^{DRP}(x_r, x_b, x_t, h_m, \lambda') = 0 \tag{6}$$

**Theorem 2** By invoking (6) and $Lemma D.1$ in [17], the distance between neural network approximation $h$ and PDE solution $u^*$ is bounded by:

$$\|u^* - h\|_{C^0(\Omega)} \leq C(\|f - \mathcal{L}[h]\|_{C^0(\Omega)} + \|g - h\|_{C^0(\partial\Omega)}) \tag{7}$$

for any $h \in C^0(\Omega)$ satisfying $\mathcal{L}[h] \in C^{0,\alpha}(\Omega)$ and $h \in C^{0,\alpha}(\partial\Omega)$. Notice that the two terms on the right hand side converge to zero for $h_m$ as PINN loss converges to 0. Thus, all lemmas and theorems above together indicate that one can learn a neural network $h$ that well approximates the true PDE solution $u$, even with misspecification, for large $m_t$ by choosing appropriate $\lambda$. In practice, the value of $\lambda$ can be set using cross-validation, as we discuss in next section, to trade off amount of misspecification with amount of data.

## Turning Machine Data Analysis

The lathe machine is prone to failure during the tuning process, and thus the experts have been seeking the help of machine learning to detect the chatter and causes of the chatter [6]. The real system data are samples from experiments performed by [22], where sensors were used to measure the operation of the lathe machine during the turning process. The

data consisted of accelerometer and audiosensor recordings of lathe machines during a turning process under multiple experimental settings (details in Appendix B). While previous literature has shown the general ODE governing machine tuning [16]: $a\frac{d^2}{dt^2}y(t) + b\frac{d}{dt}y(t) + cy(t) + d = 0$ , the details of parameters (the $a, b, c, d$ in the ODE form) are dependent on the configurations of lathe machines.

**Expert's Estimation**  We have recruited nine participants to give an estimation of the ODEs related to one damped harmonic oscillator. The details of the experiment settings and experts' prediction can be found in Appendix B. We measure the mean squared error of the solution of the experts with the collected data and see that even good experts give a MSE of 0.079 on the normalized scale, which shows the deviation of the expert's prediction from the actual system.

**PINN's Estimation Under Misspecified Experts**  In our evaluation, we utilize the expert's ODE with least MSE and build a 6 layer DNN to optimize the data-regularized PINN loss, which is defined as $\lambda loss_{data} + loss_{PINN}$, where $\lambda$ is the regularization coefficient (notice using $\lambda$ to regularize either $loss_{data}$ or $loss_{PINN}$ are equivalent, in theory part we add $\lambda$ to $loss_{PINN}$, and here we add $\lambda$ to $loss_{data}$, hence the $\lambda$ here increases to emphasize more reliance on data). We set $\lambda > 1$ to rely more on measurement data than misspecified physics. For comparison, we also trained another network that minimizes data loss only, with all other setting identical as previous. The prediction result is shown in Figure 1.

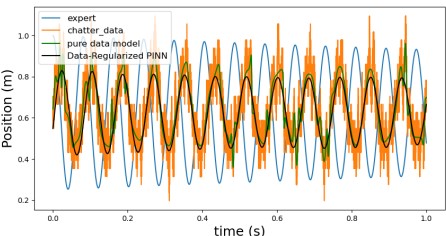

Figure 1: Prediction of combined PINN and pure data model, PINN ($MSE_{data}$=6.3e-3, $MSE_{ODE}$=1.7e-3) shows smooth result while data model ($MSE_{data}$=4.2e-3, $MSE_{ODE}$=6.4) causes heavy overfitting.

It is obvious that the data-regularized PINN results in a smoother and more rational result than pure data-driven network, which overfits the noisy data and cannot represent the underlying physics. Besides, we invoke a cross-validation model selection approach to choose best $\lambda$ that gives the optimal $\lambda$ choice, detailed in Appendix C.

## Imbalanced Data Distribution

Here we focus on another practical problem in utilizing PINN: what if we could only obtain measurement from partial domain? This is particularly true in physics, as data is not available when temperature is high in a diffusion system, or velocity is high in a dynamic motion system, etc. Fig 2 shows that, for periodic functions, vanilla PINN approach can only work well within domains that have data, but converges to an arbitrary solution in non-measurable domains. We elaborate on this aspect further in the next section.

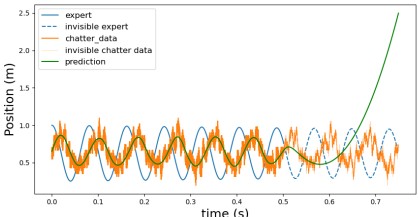

Figure 2: Vanilla PINN result over chatter data, solid/dash lines represent visible/invisible data and physics to network.

To resolve the scenario when only partial data is available, we leverage the advantage of LSTM in predicting sequential data or time-continuous functions. We first build a vanilla PINN with data regularization to well approximate the solution values in the domain where data is available, then the model is followed by a convolutional LSTM, trained with the prediction from PINN, to predict next labels in the domain where we have no data. Since the prediction from PINN has contained enough physics information, the metric when learning the ConvLSTM is simply the means square error between prediction and labels, without any physics-informed regularization.

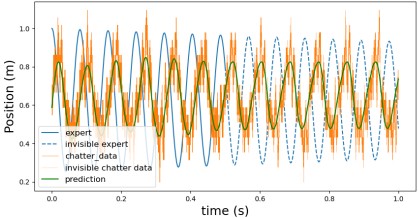

Figure 3: DNN+LSTM prediction over chatter data, noticing information of second-half is invisible to neural networks.

The evaluation of proposed method is illustrated by Figure 3, which indicates that the proposed method conducts better and more rational predictions than vanilla PINN. Though the result can be much dependent on the context length of the ConvLSTM, our empirical knowledge indicates setting context length to approximately equal to the period length leads to smooth and reasonable predictions.

## Failure of PINN: Analysis and Solution

Despite the exciting achievements of PINN, under certain physical circumstances, it also exhibits multiple optimization difficulties. The most severe one is that, without measurement data, PINN fails to approximate periodic physics solutions with high frequency and equivalently, PINN displays difficulties in converging to a global optimal when one is trying to approximate too many periods by a single neural network. Similar findings were also reported by [7]. However, no existing work explains the optimization landscape that leads to this phenomenon.

To understand why PINN fails to approximate high frequency functions, we start with a toy spring mass system,

which has general formulation: $mu_{tt} + ku = 0, u(0) = A, u_t(0) = B$, where $u_{tt}$ is equivalent to $\frac{d^2}{dt^2}u(t)$.

A general solution to this ODE system under given initial condition (IC) and boundary condition (BC) is $u(t) = A\cos\omega t + \frac{B}{\omega}\sin\omega t$, where $\omega = \sqrt{\frac{k}{m}}$. Nevertheless, ignoring the IC and BC, a trivial solution to the system is $u(t) = 0$. Though $u(t) = 0$ is analytically incorrect solution to the given system given the initial and boundary conditions, we should recognize that the neural network is a numerical approach with a black box operator. Recall that vanilla PINN defines an optimization objective as $loss_{PINN} = loss_{PDE} + loss_{BC+IC}$. When the collocation points are close enough to the initial state, $loss_{BC+IC}$ plays constraints in approximation. However, as collocation points become away from initial state, $loss_{BC+IC}$ will no longer be a constraint to those points, and the neural network will approximate arbitrary solutions that fit the physics equation, including the trivial solution. Thus, PINN intends to learn a combined solution with correct solution in the beginning, and an arbitrary solution, typically a trivial solution due to its simplicity to approximate, in the later part of the domain, which is analytically incorrect but numerically possible. This phenomenon for learning a simple spring mass system with PINN is shown in Figure 4.

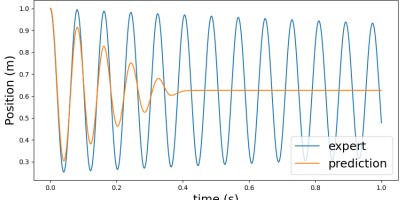

Figure 4: PINN solver of spring mass system, which underfits the true solution because $loss_{BC+IC}$ plays limited effect.

This phenomenon is caused due to imbalanced optimizations in different parts of the domain. In Figure 4, we observe a 'fast decay' of approximation for $t \leq 0.4$ and eventually it converges to a trivial solution. For clarity, we name part $t \leq 0.4$ as *decaying region*, and the remaining part as *flat region*. It turns that the ODE loss is small in the flat region, but not in the decaying region. Since PINN optimizes the ODE loss over the whole domain and the trivial solution is easy to approximate, if too many collocation points are sufficient periods away from initial state, then optimizing these points to a trivial solution is more favorable. Since a trivial solution is well optimized (error is 0 or close to 0) at the flat region, its gradient can be extremely trivial such that the neural network can never jump from the local optima.

In [7], Seq2seq training was proposed as a fix for this problem. Seq2seq trains multiple PINNs by slicing the domain sequentially, in which the last prediction from previous PINN serves as the initial condition for the next PINN. Through leveraging seq2seq training, one could slice the domain into parts, in which each part has much fewer periods than original domain, possibly make more collocation points close to the initial state to avoid estimating a flat region.

One problem for seq2seq training is that the use of ini-

tial condition from prediction makes error cumulative, leading to an inferior approximation in later part of the domain. In addition, empirical evaluation indicates seq2seq training with vanilla PINN is typically slow in convergence and hard to reach a desired error. Thus our works make following adaptions for better convergence: (1) Inspired by [8], instead of softly enforcing the initial conditions, we try to hard-code the initial conditions using second-order Taylor approximation;(2) Instead of directly learning with a given equation, we stretch the equation in time domain to a lower frequency, and squeeze its prediction back for normal approximation.

To illustrate the proposed modification, we leverage a simple ode $au_{tt} + bu_t + cu + d = 0$, with IC/BC $u(0) = A, u_t(0) = B$, which is a variant of a spring mass system. The first trick indicates an approach of approximating:

$$u = A + Bt + t^2h \qquad (8)$$

rather than $u = h$. Here $h$ refers to the neural network, which makes it harder for the neural network to approximate a trivial solution. The second trick is that we stretch $t$ by k times and evaluate PINN loss based on:

$$k^2au_{tt} + kbu_t + cu + d = 0 \qquad (9)$$

whose solution is in same shape, but different in time scale, we then let $u'(t) = u(kt)$ to reproduce the original solution.

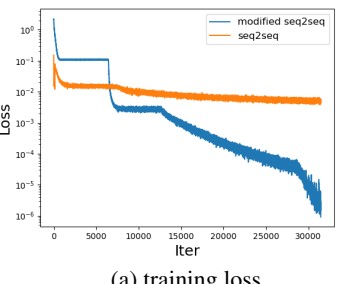

(a) training loss

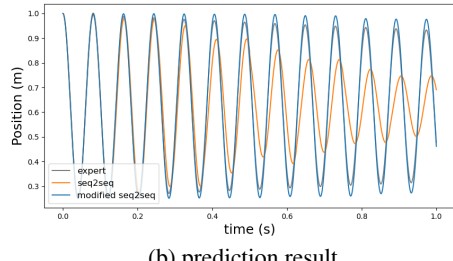

(b) prediction result

Figure 5: Training loss and prediction for seq2seq and modified-seq2seq, with no measurement data.

By letting $a = 0.8, b = 0.32. c = 4800, d = -3000$ and $A = 1, B = 0$, we empirically evaluate the proposed modifications versus original seq2seq training. Figure 5 shows a faster convergence of error for modified seq2seq, when both models are optimized using Adam optimizer with $lr = 0.001$, indicating the proposed method is easier for training. Since vanilla seq2seq training cannot converge to a desired loss, it leads to a larger error for next initial state. Cumulatively, the error will be magnified through cascade, and eventually result in an inferior prediction.

## Conclusion

In this paper we have theoretically proven that PINN can recover the ordinary and partial differential equation with limited data, even when the governing physics knowledge that enforces PINN's loss function is not a precise estimation. We further demonstrate the effectiveness of PINN with a real-life scenario of machine turning process and predictive maintenance. We have also analyzed the potential failure characters of PINN because of its difficulty to converge to a global optimal or the lack of data throughout the input domain. We have provided the modified PINN models to target each of the above scenarios.

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

# Appendix
## A. Proof of Lemma 1.1 and Theorem 1
*Proof of Lemma 1.1*

Instead of considering a special case of Dirichlet boundary condition, let start with a general PDE with misspecification:

$$(\mathcal{L} + \epsilon_{\mathcal{L}})u(x) = f(x) + \epsilon_f(x), \ \forall x \in \Omega$$
$$(\mathcal{B} + \epsilon_{\mathcal{B}})u(x) = g(x) + \epsilon_g(x), \ \forall x \in \partial\Omega$$

The Misspecified Data-Regularized PINN Loss and its empirical loss are modified accordingly:

$$Loss^{MDRP}(x_r, x_b, x_t, h, \gamma) = \|h(x_t) - u(x_t)\|^2$$
$$+ \lambda_r \|(\mathcal{L} + \epsilon_{\mathcal{L}})[h](x_r) - f(x_r) - \epsilon_f(x_r)\|^2$$
$$+ \lambda_b \|(\mathcal{B} + \epsilon_{\mathcal{B}})[h](x_b) - g(x_b) - \epsilon_g(x_b)\|^2$$

$$Loss_m^{MDRP}(h, \gamma) = \frac{1}{m_t} \sum_{i=1}^{m_t} \|h(x_t^i) - u(x_t^i)\|^2$$

$$+ \frac{\lambda_r}{m_r} \sum_{i=1}^{m_r} \|(\mathcal{L} + \epsilon_{\mathcal{L}})[h](x_r^i) - f(x_r^i) - \epsilon_f(x_r^i)\|^2$$

$$+ \frac{\lambda_b}{m_b} \sum_{i=1}^{m_b} \|(\mathcal{B} + \epsilon_{\mathcal{B}})[h](x_b^i) - g(x_b^i) - \epsilon_g(x_b^i)\|^2$$

Invoking the inequality: $\|x + y + z\|^2 \leq \|x\|^2 + \|y\|^2 + \|z\|^2$, we have:

$$\|\mathcal{L}[h](x_r) - f(x_r)\|^2 \leq 3(\|\mathcal{L}[h](x_r') - f(x_r')\|^2$$
$$+ \|\mathcal{L}[h](x_r) - \mathcal{L}[h](x_r')\|^2 + \|f(x_r) - f(x_r')\|^2)$$
$$\|\mathcal{B}[h](x_b) - f(x_b)\|^2 \leq 3(\|\mathcal{B}[h](x_b') - g(x_b')\|^2$$
$$+ \|\mathcal{B}[h](x_b) - \mathcal{B}[h](x_b')\|^2 + \|g(x_b) - g(x_b')\|^2)$$

Again, apply $\|x + y + z\|^2 \leq \|x\|^2 + \|y\|^2 + \|z\|^2$ to $\|\mathcal{L}[h](x_r') - f(x_r')\|^2$ and $\|\mathcal{B}[h](x_b') - g(x_b')\|^2$, we have:

$$\|\mathcal{L}[h](x_r') - f(x_r')\|^2 \leq 3(\|(\mathcal{L} + \epsilon_{\mathcal{L}})[h](x_r') - f(x_r')$$
$$- \epsilon_f(x_r')\|^2 + \|\epsilon_{\mathcal{L}}[h](x_r')\|^2 + \|\epsilon_f(x_r')\|^2)$$
$$\|\mathcal{B}[h](x_b') - g(x_b')\|^2 \leq 3(\|(\mathcal{B} + \epsilon_{\mathcal{B}})[h](x_b') - g(x_b')$$
$$- \epsilon_g(x_b')\|^2 + \|\epsilon_{\mathcal{B}}[h](x_b')\|^2 + \|\epsilon_g(x_b')\|^2)$$

In addition, we derive the measurement data term as:

$$\|h(x_t) - u(x_t)\|^2 \leq 3(\|h(x_t') - u(x_t')\|^2$$
$$+ \|h(x_t) - h(x_t')\|^2 + \|u(x_t') - u(x_t)\|^2)$$

With all inequalities listed above, we have:

$$Loss^{DRP}(x_r, x_b, x_t, h, \lambda) \leq 3\|h(x_t') - u(x_t')\|^2 + 3\|h(x_t)$$
$$- h(x_t')\|^2 + 3\|u(x_t') - u(x_t)\|^2 + 9\lambda_r(\|(\mathcal{L} + \epsilon_{\mathcal{L}})[h](x_r')$$
$$- f(x_r') - \epsilon_f(x_r')\|^2 + \|\epsilon_{\mathcal{L}}[h](x_r')\|^2 + \|\epsilon_f(x_r')\|^2)$$
$$+ 9\lambda_b(\|(\mathcal{B} + \epsilon_{\mathcal{B}})[h](x_b') - g(x_b') - \epsilon_g(x_b')\|^2$$
$$+ \|\epsilon_{\mathcal{B}}[h](x_b')\|^2 + \|\epsilon_g(x_b')\|^2) + 3\lambda_r(\|\mathcal{L}[h](x_r)$$
$$- \mathcal{L}[h](x_r')\|^2 + \|f(x_r) - f(x_r')\|^2) + 3\lambda_b(\|\mathcal{B}[h](x_b)$$
$$- \mathcal{B}[h](x_b')\|^2 + \|g(x_b) - g(x_b')\|^2)$$

By letting $\lambda' = 3\lambda$, and leveraging the property of Hölder space, we could bound $Loss^{DRP}$ by $Loss^{MDRP}$ by:

$$Loss^{DRP}(x_r, x_b, x_t, h, \lambda) \leq 3Loss^{MDRP}(x_r', x_b', x_t', h, \lambda')$$
$$+ 3\epsilon_t^{2\alpha}[h]_{\alpha,\Omega}^2 + 3\epsilon_t^{2\alpha}[u]_{\alpha,\Omega}^2 + \lambda_r'\epsilon_r^{2\alpha}[\mathcal{L}[h]]_{\alpha,\Omega}^2$$
$$+ \lambda_r'\epsilon_r^{2\alpha}[f]_{\alpha,\Omega}^2 + \lambda_b'\epsilon_b^{2\alpha}[\mathcal{B}[h]]_{\alpha,\partial\Omega}^2 + \lambda_b'\epsilon_b^{2\alpha}[g]_{\alpha,\partial\Omega}^2$$
$$+ 3\lambda_r'(\|\epsilon_{\mathcal{L}}[h](x_r')\|^2 + \|\epsilon_f[h](x_r')\|^2)$$
$$+ 3\lambda_b'(\|\epsilon_{\mathcal{B}}[h](x_b')\|^2 + \|\epsilon_g[h](x_b')\|^2)$$

For writing simplicity, we use $\epsilon_h$ to represent the error terms with Hölder constants and $\epsilon_m$ to represent misspecification error terms. Assuming there exists positive constants $c_r, c_b, c_t$, such that $\forall \epsilon > 0$, $A_{x_r^i}$, $A_{x_b^i}$, and $A_{x_t^i}$ satisfy $c_r \epsilon^d \leq \mu_r(A_{x_r^i})$, $c_b \epsilon^{d-1} \leq \mu_b(A_{x_b^i})$, and $c_t \epsilon^d \leq \mu_t(A_{x_t^i})$. Also there exists positive constants $C_r, C_b, C_t$, s.t. $\forall x_r, x_t \in \Omega$, $x_b \in \partial\Omega$, such that $\mu_r(B_\epsilon(x_r) \cap \Omega) \leq C_r \epsilon^d$, $\mu_b(B_\epsilon(x_b) \cap \partial\Omega) \leq C_b \epsilon^{d-1}$, and $\mu_t(B_\epsilon(x_t) \cap \Omega) \leq C_t \epsilon^d$. Here, $A_{x_r^i}$, $A_{x_b^i}$, and $A_{x_t^i}$ are the Voronoi cell defined as:

$$A_{x_r^i} = \{x \in U | \|x - x_r^i\| = \min_{x' \in \tau_r} \|x - x'\|\}$$
$$A_{x_b^i} = \{x \in \Gamma | \|x - x_b^i\| = \min_{x' \in \tau_b} \|x - x'\|\}$$
$$A_{x_t^i} = \{x \in U | \|x - x_t^i\| = \min_{x' \in \tau_t} \|x - x'\|\}$$

Let $\gamma_r^i = \mu_r(A_{x_r^i})$, $\gamma_b^i = \mu_r(A_{x_b^i})$, and $\gamma_t^i = \mu_t(A_{x_t^i})$, let $\gamma_r^* = \max_i \gamma_r^i$, $\gamma_b^* = \max_i \gamma_b^i$, and $\gamma_t^* = \max_i \gamma_t^i$, by taking expectations with respect to $(x_r, x_b) \sim \mu = \mu_r \times \mu_b$, we could express the expectation in terms of empirical loss as:

$$\mathbf{E}_\mu[L(x_r, x_b, h, \lambda)]$$
$$\leq 3m_r \gamma_r^* \frac{\lambda_r'}{m_r} \sum_{i=1}^{m_r} \|(\mathcal{L} + \epsilon_{\mathcal{L}})[h](x_r^i) - f(x_r^i) - \epsilon_f(x_r^i)\|^2$$
$$+ 3m_b \gamma_b^* \frac{\lambda_b'}{m_b} \sum_{i=1}^{m_b} \|(\mathcal{B} + \epsilon_{\mathcal{B}})[h](x_b^i) - g(x_b^i) - \epsilon_g(x_b^i)\|^2$$
$$+ 3m_t \gamma_t^* \frac{1}{m_t} \sum_{i=1}^{m_t} \|h(x_t^i) - u(x_t^i)\|^2 + \epsilon_h + \epsilon_m$$

where $L(x_r, x_b, h, \lambda) = (\lambda_r \|\mathcal{L}[h](x_r) - f(x_r)\|^2)\mathbf{I}_\Omega(x_r) + (\lambda_b \|\mathcal{B}[h](x_b) - g(x_b)\|^2)\mathbf{I}_{\partial\Omega}(x_b)$, and $\mathbf{I}_A(x)$ is an indicator function on set $A$. Based on the assumption above, we also have $\gamma_r^* \leq C_r \epsilon_r^d$, $\gamma_b^* \leq C_b \epsilon_b^{d-1}$, and $\gamma_t^* \leq C_t \epsilon_t^d$. Then we can obtain:

$$\mathbf{E}_\mu[L(x_r, x_b, h, \lambda)]$$
$$\leq 3C_r m_r \epsilon_r^d \frac{\lambda_r'}{m_r} \sum_{i=1}^{m_r} \|(\mathcal{L} + \epsilon_{\mathcal{L}})[h](x_r^i) - f(x_r^i) - \epsilon_f(x_r^i)\|^2$$
$$+ 3C_b m_b \epsilon_b^{d-1} \frac{\lambda_b'}{m_b} \sum_{i=1}^{m_b} \|(\mathcal{B} + \epsilon_{\mathcal{B}})[h](x_b^i) - g(x_b^i) - \epsilon_g(x_b^i)\|^2$$
$$+ 3C_t m_t \epsilon_t^d \frac{1}{m_t} \sum_{i=1}^{m_t} \|h(x_t^i) - u(x_t^i)\|^2 + \epsilon_h + \epsilon_m$$
$$\leq C_m' Loss_m^{MDRP}(h, \lambda') + \epsilon_h + \epsilon_m$$

where $C'_m = 3 \max\{C_r m_r \epsilon_r^d, C_b m_b \epsilon_b^{d-1}, C_t m_t \epsilon_t^d\}$. Then the proof is completed.

*Proof of Theorem 1*

By expanding $\epsilon_h$ and $\epsilon_m$, let $\epsilon_{mr} = 3(\|\epsilon_{\mathcal{L}}[h](x'_r)\|^2 + \|\epsilon_f[h](x'_r)\|^2)$ and $\epsilon_{mb} = 3(\|\epsilon_{\mathcal{B}}[h](x'_b)\|^2 + \|\epsilon_g[h](x'_b)\|^2)$ to denote the misspecified errors. Also let $\epsilon_{hr} = [\mathcal{L}[h]]^2_{\alpha,\Omega} + [f]_{\alpha,\Omega}$, $\epsilon_{hb} = [\mathcal{B}[h]]^2_{\alpha,\partial\Omega} + [g]_{\alpha,\partial\Omega}$, and $\epsilon_{ht} = 3([h]^2_{\alpha,\Omega} + [u]_{\alpha,\Omega})$ to denote the errors involved with Holder constants. That is, $\epsilon_h + \epsilon_m = \lambda'_r(\epsilon_r^{2\alpha}\epsilon_{hr} + \epsilon_{mr}) + \lambda'_b(\epsilon_b^{2\alpha}\epsilon_{hb} + \epsilon_{mb}) + \epsilon_t^{2\alpha}\epsilon_{ht}$. By leveraging *Lemma 1.2*, rather than setting $\epsilon_r = c_r^{-\frac{1}{d}} m_r^{-\frac{1}{2d}}$, $\epsilon_b = c_r^{-\frac{1}{d-1}} m_r^{-\frac{1}{2(d-1)}}$, we tighten the bound to $\epsilon_r = (\frac{2\log m_r}{c_r m_r})^{\frac{1}{d}}$, $\epsilon_b = (\frac{2\log m_b}{c_b m_b})^{\frac{1}{d-1}}$, and similarly, we set $\epsilon_t = (\frac{2\log m_t}{c_t m_t})^{\frac{1}{d}}$. Then the following bound holds with $> 1 - \delta$ probability for any $\delta \in (0,1)$. with probability. The Data-Regularized PINN Loss can be bounded as:

$$Loss^{PINN}(x_r, x_b, x_t, h, \lambda) \leq Loss^{DRP}(x_r, x_b, x_t, h, \lambda)$$

$$\leq C'_m Loss_m^{MDRP}(h, \lambda') + \lambda'_r((\frac{2\log m_r}{c_r m_r})^{\frac{2\alpha}{d}}\epsilon_{hr} + \epsilon_{mr})$$

$$+ \lambda'_b((\frac{2\log m_b}{c_b m_b})^{\frac{2\alpha}{d-1}}\epsilon_{hb} + \epsilon_{mb}) + (\frac{2\log m_t}{c_t m_t})^{\frac{2\alpha}{d}}\epsilon_{ht}$$

By setting $\lambda'_r = \lambda'_b = \frac{1}{m_t}$, we could conclude the empirical loss will converge to 0 as $m_t \to \infty$. Suppose $m_r = \mathcal{O}(m_t)$ and $m_b = \mathcal{O}(m_t^{\frac{d-1}{d}})$, since we assume that there exists $h_m \in \mathcal{H}$ such that $h_m$ perfectly optimizes the empirical data-regularized PINN loss, that is, $Loss_m^{MDRP}(h_m, \frac{1}{m_t}) = 0$, then the upper bound becomes:

$$Loss^{DRP}(x_r, x_b, x_t, h, \lambda) \leq \mathcal{O}(m_t^{-1}) + \mathcal{O}(m_t^{-(\frac{2\alpha}{d}+1)} \log m_t)$$

$$+ \mathcal{O}(m_t^{-\frac{2\alpha}{d}} \log m_t)$$

Then the proof is completed.

In addition, for picking proper $\lambda_r, \lambda_b$, rather than simply setting to $\frac{1}{m_t}$, [2] proposed a method that automatically choosing appropriate $\lambda$ by hold-out validation.

## B. Expert's Evaluation of Lathe Machine Tuning

Here we summarized our experiment and the results we have acquired from the experts.

### Participants

Participants were eligible for the study if they had a mechanical engineering background with a focus on machining theory and operation of lathe machines. Our recruitment efforts focused on: 1) researchers with expertise in machining dynamics, 2) undergraduate and graduate mechanical engineering students who have training in machining theory and operation of lathe machines, and 3) technicians who supervise and maintain lathe machines. Participants were recruited by online free-lancing work platforms (Upwork, Fiverr). Experts were also asked to complete a preliminary survey asking about the years of experience and domain specializations. We recruited nine individuals from June 30, 2021 to August 30, 2021 including one researcher, six technicians/engineers, and two students.

### Procedure

Experts were asked to provide equations for a lathe machine under normal operation. In the tasks, the experts were asked to provide judgments about a simulated system with a data generating process known to the experimenters, but hidden to the participant. While the machine turning process generally follows a second-order ODE as:

$$a\frac{d^2}{dt^2}y(t) + b\frac{d}{dt}y(t) + cy(t) + d = 0$$

we would like the experts to give an estimation of the parameters in the ODE.

We utilize the normal operation part of F_12-Jun-2017_rpm570_doc0p008.mat collected by [22]. We observe the turning machine data is 16000 points from interval $t \in [0, 0.1]$ with a sampling frequency of $6.2e^{-6}$. Since the original data space is relatively too small for neural network training, we normalize the data to $u(t)/|(u(t)_{max} - u(t)_{min}|$ and $t/|t_{max} - t_{min}|$. Thus, the data is re-normalized to $t \in [0, 1]$ and $u(t) \in [0, 1]$. We also re-scale the expert's evaluation accordingly.

### Results

With the experts' estimation we calculated the analytical solution to the ODE problem:

$$y(t) = C_1 e^{\frac{t(-b-\sqrt{-4ac+b^2})}{2a}} + C_2 e^{\frac{t(-b+\sqrt{-4ac+b^2})}{2a}} - \frac{d}{c}$$

With $C_1$ and $C_2$ are calculated based on each individual's parameters. We take the same sampling frequency on the analytical solutions and measure the mean squared distance to the actual noisy data. The rescaled MSE is calculated by averaging $L2$-distance the between rescaled expert's prediction of $16,000$ time points and normalized data.

Table 1: The results shown by nine experts and we measure the scaled MSE to the normalized turning machine data.

| Expert | Estimated Values | | | | Scaled MSE |
|---|---|---|---|---|---|
| | a | b | c | d | |
| 1 | 1 | 4 | 600,000 | -3000 | 0.120 |
| 2 | 1 | 0 | 450,000 | -2500 | 0.079 |
| 3 | 1e-7 | 0 | 0.04 | 0.0025 | 0.89 |
| 4 | 1 | 0 | 650,000 | -2500 | 0.199 |
| 5 | 1 | 0 | 500,000 | -2500 | 0.081 |
| 6 | 6 | 0 | 2,627,000 | 0 | 0.489 |
| 7 | 1 | 0 | 10,000,000 | -50000 | 0.096 |
| 8 | 1 | 0 | 322,000 | -1720 | 0.124 |
| 9 | 20 | 0.5 | 10,000,000 | -20000 | 0.425 |

## C. PINN Result with Cross-validation

Here we apply Expert 1's prediction as the prior physics information fed into PINN. And to illustrate the effect of regularization on data, we assume the initial and boundary conditions based on chatter data are exactly given to PINN, so PINN does not need to learn the additional parameters. We also perform a normalization of related ODE,

such that the largest coefficient in ODE is normalized to 1, and PINN could learn the solution with fewer iterations, and the learning rate can be enlarged to allow faster searching of the optimization space. To achieve this, we apply a Taylor-expansion based approach to decompose the output of PINN into:

$$y(t) = A(1 - t) + Bt + t(1 - t)h$$

Where $A = y(0)$ and $B = y(1)$ are the normalized boundary conditions over $t$, intuitively, we hard-coded the two points boundary conditions to easier training. The $h$ is the output of PINN when minimizing the data-regularized loss: $\lambda loss_{data} + loss_{PINN}$. For the neural network we apply a six-layer neural network with number of nodes at $[1, 64, 64, 64, 10, 1]$ at each layer, and uniformly sampled 8000 out of the 16000 points from either data or collocation points for training, and rest points for validation. For all models we trained, the training index and validation index remain identical.

In the table 2 we provide a brief summary of the effects of regularization on data loss and ODE loss computed on validation set. We observe that as the regularization gets larger, there is an increase in ODE loss because the PINN relies more on data rather than the inaccurate physics prior. Similarly the data loss is decreasing because the PINN is more fitted to the data. However, we also notice that just increasing the regularization to a significantly large amount is not realizable: the data loss starts increasing at certain level when the model starts overfitting to data. Interestingly, the data loss suddenly decreases at extreme large regularizations, however this is still overfitting as the prediction actually fails to follow physics rules as is reflected in large ODE loss (see Figure 7).

Thus, the effectiveness of cross-validation can be interpreted as follows: First, we expect the chattering data follows some physics rules, and a large $\lambda$ which relies too heavily on data causes large physics error, making prediction less reliable; Second, the noise of data might not be iid or zero mean, such that there exists biases in data MSE measurement (and such correlations in noise between training and validation set might explain why the model overfits to validation data for very large $\lambda$). In the table, we highlight two results and plot the results of PINN with experts' prediction and noisy data. We notice that even through PINN solution, shown with Figure 7, is at larger regularization and provides smaller data loss, its solution tends to overfit to the data provided and is less informative of the general physics knowledge. The PINN solution with regularization $\lambda = 1$, shown in Figure 6 fits the general trend better even though it has a slightly larger data loss.

However, no matter the regularization, we notice that in general the MSE of the data+physics based model is smaller than the experts' predictions (physics only model, where the expert provided physics is misspecified), which shows how PINN with combination of experts and data can improve our understanding of the machinery processes. Thus, this hybrid method can flexibly combine expert physical knowledge with data-driven models making it possible to obtain the best of both worlds: small-data accuracy by leveraging

expert judgement and large-data generalizability despite potentially misspecified expert knowledge.

Table 2: Cross validation result. As regularization becomes larger, we observe an increase in ODE loss and decrease in Data loss.

| Regularization | Data MSE | ODE MSE |
|---|---|---|
| 1e-10 | 2.34e-2 | 7.19e-3 |
| 1e-5 | 1.97e-2 | 4.51e-6 |
| 1e-3 | 2.23e-2 | 4.80e-5 |
| 1e-1 | 9.61e-3 | 2.12e-4 |
| 1e+0 | 6.36e-3 | 3.64e-4 |
| 1e+1 | **6.31e-3** | 2.43e-3 |
| 1e+3 | 5.50e-3 | 1.99e-1 |
| 1e+5 | 5.43e-3 | 2.22e+0 |
| 1e+7 | 8.41e-3 | 1.23e+1 |
| 1e+9 | 1.90e-2 | 4.06e+1 |
| 1e+12 | **4.27e-3** | 1.75e+1 |
| 1e+15 | 5.18e-3 | 1.33e+1 |

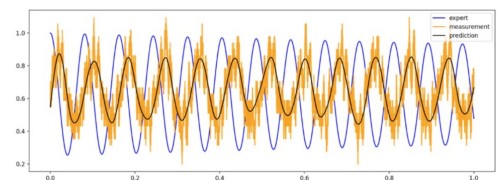

Figure 6: Regularization is 1e+1. The PINN prediction captures the general trend.

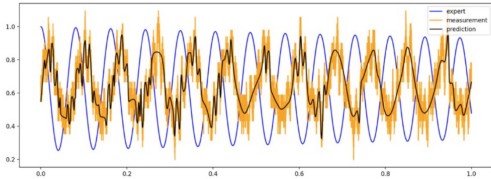

Figure 7: Regularization is 1e+12. The PINN prediction tends to overfit to the data, though it leads to lower MSE error on validation data.