# OpenReview forum: "Physics Informed Machine Learning with Misspecified Priors: \\An analysis of Turning Operation in Lathe Machines"
_AAAI.org/2022/Workshop/ADAM — AAAI 2022 Workshop ADAM_

### Official Review · Reviewer_tviE · 2021-11-29
**Well written paper; minor clarification required**

**Rating:** 8
**Confidence:** 4

**Review:**

This paper has two major contributions: (1) Providing the theoretical convergence proofs of PINNs under misspecification, and (2) using PINNs, tackling the problem of predictive maintenance of Lathe machines, specifically under imbalances data distribution. Both the contributions are novel and relevant to this workshop. The paper is well written and the results are good and the theoretical proof is sound. Here are a few questions for some minor clarification.

1. While Theorem 1 is sound and is justified based on the assumptions, I am not sure as a machining expert if I want the Loss to be zero. In other words, as a practitioner of deep learning in manufacturing, I would be interested in seeing a term bounded by \epsilon (the iid noise of the empirical data) in Equation 5. That could help me in controlling the noise in the data collected.

2. In the results section are the authors letting the *a,b,c,d* of the ODE be fixed during training PINNs or they are set to be *trainable*? If all the expert is doing is to minimize MSE by playing around with *a,b,c,d*, it might be better to make a,b,c, and d be *trainable* and thus the authors may get better performance.

3. The term pure data model, PINN is confusing in figure 1. PINN (traditionally are known as *data-free* methods) cannot be called a pure data model.

4. This idea of imbalance data distribution is similar to that pursued in this paper: https://ojs.aaai.org/index.php/AAAI/article/view/16992 (although called as extrapolation capability than imbalanced dataset). The authors may benefit from the probable comparison with the method they propose here.

overall, the paper looks good. The above-mentioned clarifications would be good.

---

### Official Review · Reviewer_qsnc · 2021-12-01
**Analysis of PINNs under misspecified priors**

**Rating:** 9
**Confidence:** 5

**Review:**

The paper presents an analysis of PINNs under misspecified priors. The authors prove that PINNs converge even under these situations. The authors then demonstrate a practical application of the proposed idea on a time-series problem of lathe chatter.

The paper is well written, and the given theorems prove that the PINNs converge even with misspecified priors. I had a minor comment regarding the assumption of ellipticity for the given ODEs. I was not sure where this was used to prove the convergence. Would PINNs also converge for misspecified priors, even for other kinds of ODEs?

Overall, this is a great workshop paper.